# Translational research into the effects of cigarette smoke on inflammatory mediators and epithelial TRPV1 in Crohn's disease

**Liesbeth Allais**[1,2]*, **Stephanie Verschuere**[3], **Tania Maes**[4], **Rebecca De Smet**[2], **Sarah Devriese**[5], **Gerard Bryan Gonzales**[5], **Harald Peeters**[6], **Koen Van Crombruggen**[7], **Claus Bachert**[7], **Martine De Vos**[5], **Guy G. Brusselle**[4], **Ken R. Bracke**[4], **Claude A. Cuvelier**[2ᵒ], **Debby Laukens**[5ᵒ]

**1** Animal Healthcare Flanders, Torhout, Belgium, **2** Department of Medical and Forensic Pathology, Ghent University, Ghent, Belgium, **3** Department of Pathology, AZ Delta, Roeselare, Belgium, **4** Department of Respiratory Medicine, Laboratory for Translational Research in Obstructive Pulmonary diseases, Ghent University Hospital, Ghent, Belgium, **5** Department of Gastroenterology, Ghent University, Ghent, Belgium, **6** Department of Gastroenterology, AZ Sint-Lucas, Ghent, Belgium, **7** Upper Airways Research Laboratory, Department of Oto-Rhino-Laryngology, Ghent University, Ghent, Belgium

ᵒ These authors contributed equally to this work.

\* liesbeth.allais@gmail.com

**Data Availability Statement:** All relevant data are within the manuscript and its Supporting Information files.

## Abstract

Crohn's disease is a pathological condition of the gastro-intestinal tract, causing severe transmural inflammation in the ileum and/or colon. Cigarette smoking is one of the best known environmental risk factors for the development of Crohn's disease. Nevertheless, very little is known about the effect of prolonged cigarette smoke exposure on inflammatory modulators in the gut. We examined the effect of cigarette smoke on cytokine profiles in the healthy and inflamed gut of human subjects and in the trinitrobenzene sulphonic acid mouse model, which mimics distal Crohn-like colitis. In addition, the effect of cigarette smoke on epithelial expression of transient receptor potential channels and their concurrent increase with cigarette smoke-augmented cytokine production was investigated. Active smoking was associated with increased *IL-8* transcription in ileum of controls (p < 0,001; n = 18-20/ group). In the ileum, TRPV1 mRNA levels were decreased in never smoking Crohn's disease patients compared to healthy subjects (p <0,001; n = 20/group). In the colon, TRPV1 mRNA levels were decreased (p = 0,046) in smoking healthy controls (n = 20/group). Likewise, healthy mice chronically exposed to cigarette smoke (n = 10/group) showed elevated ileal *Cxcl2* (p = 0,0075) and colonic *Kc* mRNA levels (p = 0,0186), whereas TRPV1 mRNA and protein levels were elevated in the ileum (p = 0,0315). Although cigarette smoke exposure prior to trinitrobenzene sulphonic acid administration did not alter disease activity, increased pro-inflammatory cytokine production was observed in the distal colon (Kc: p = 0,0273; Cxcl2: p = 0,104; Il1-β: p = 0,0796), in parallel with the increase of *Trpv1* mRNA (p < 0,001). We infer that CS affects pro-inflammatory cytokine expression in healthy and inflamed gut, and that the simultaneous modulation of TRPV1 may point to a potential involvement of TRPV1 in cigarette smoke-induced production of inflammatory mediators.

**Funding:** RDS was supported by Concerted Research Action of Ghent University (BOF10/GOA/021, BOF12/GOA/008), TM and KB were supported by FWO Flanders and Interuniversitary Attraction Poles Programme/Belgian Federal Science Policy P7/30. LA was supported by a doctoral grant from the Special Research Fund of Ghent University (01D41012). The funders had no role in study design, data collection and analysis, decision to publish, or preparation of the manuscript.

**Competing interests:** The authors have declared that no competing interests exist.

# Introduction

Crohn's disease (CD) is characterized by severe gastro-intestinal inflammation and results from a complex interplay between genetic and environmental factors. In CD patients, a transmural and discontinuous inflammation affects mainly the ileum and colon, however, inflammatory lesions can extend to any part of the gastrointestinal tract [1, 2]. CD is mediated by a Th1 inflammatory response at the level of the gut mucosa [3]. Increased numbers of Th17 cells and the production of Th17-related cytokines are also reported to be associated with active inflammation in CD patients [4]. Emerging evidence suggests that the development of CD in susceptible individuals is a consequence of a dysregulated dialogue between the intestinal microbiota and the immune system of the gut [5, 6]. In addition, environmental factors affect the incidence and disease course of CD. Active smoking in particular is a very prominent risk factor [7].

Cigarette smoking (CS) doubles the risk of developing CD, and has detrimental effects on its clinical course, necessitating increased need for steroids, immunosuppressive drugs, and surgery [8, 9]. The effect of CS exposure on the intestine is complex and depends on the location of inflammation. In addition, a wide range of substances are involved including nicotine, acrolein, oxygen-free radicals and carbon monoxide. CS affects the mucus layer composition, cytokine and eicosanoid production, immune cell functions, gastrointestinal motility, microvasculature and even the composition and activity of the microbiome in the gastro-intestinal tract [10–12]. We have previously shown that chronic CS exposure triggers the gut immune system through the recruitment of immune cells to the Peyer's patches, with an increase of CCL9 and a decrease of CCR6 in the ileum [13, 14]. Experimental data investigating the effects of CS in animal models of CD, e.g. the trinitrobenzene sulphonic acid (TNBS)-induced Crohn-like colitis, are inconsistent [15]. CS exposure during four days or nicotine administration has been shown to aggravate TNBS-induced colonic inflammation, whereas also an attenuating effect or no effect at all has been reported [16–21].

To date, the mechanism underlying the modulation of cytokine production by CS remains to be revealed. Transient receptor potential (TRP) channels, which act as sensors of various intra- and extracellular stimuli, including CS components, are associated with CD-related abdominal pain and could be implicated in inflammatory processes in the gut [22]. Previous work indicated the involvement of TRP channels on sensory neurons in pro-inflammatory responses of the colon, however, no consensus on the actual role of each channel has been reached [23, 24]. Only TRPV4 has been shown to be implicated in gut inflammation [25, 26]. Until today, the importance of epithelial TRP channels in gut inflammation and the potential modulation by CS has not yet been elucidated. The importance of TRP channels and CS in intestinal inflammation has been reviewed recently [27].

In this manuscript, we investigated the effect of CS exposure on CD using gut tissue of patients and a murine colitis model. We demonstrated that CS modulates pro-inflammatory mediator production and TRPV1 expression and suggest that epithelial TRPV1 activation may precede pro-inflammatory mediator release in the gut. We observed changes in IL-8 and TRPV1 in the ileum of CD patients, which prompted us to investigate whether CS-induced TRPV1 expression might be linked to a simultaneous cytokine/chemokine induction in experimental mouse models and human gut cell lines.

# Materials and methods

## Patients

In total, 155 human subjects were examined. All patients were recruited at the Ghent University Hospital between 2010 and 2015. Patients and controls with a Pack Year of at least 10 were

**Table 1. Characteristics of ileal biopsy donors.**

| | *Control Ileum* | | *Ileal CD* | |
| --- | --- | --- | --- | --- |
| | **Never smokers** | **Current smokers** | **Never smokers** | **Current smokers** |
| Number of subjects | 20 | 18 | 18 | 19 |
| Pack Year (mean, IQR) | - | 15,9 (4–50) | - | 15,9 (7,5–37) |
| Age (median, IQR) in years | 53 (28–74) | 51 (39–72) | 34 (28–76) | 43 (30–63) |
| Age at diagnosis (median, IQR) in years | - | - | 28 (22–57) | 31 (23–60) |
| Gender (Female/Male) | 13/7 | 11/7 | 10/8 | 11/8 |
| C-reactive protein (mean ± SD, IQR) in mg/dl | 1,1 ± 2,3 (0,1–9,3) | 2,4 ± 4,2 (0,1–14,8) | 6,6 ± 13,7 (0,5–52,5) | 9,9 ± 12,0 (0,9–37,1) |
| **Medication** | | | | |
| No Medication | 20 | 18 | 1 | 3 |
| Immunosuppressives | 0 | 0 | 1 | 9 |
| Corticosteroids | 0 | 0 | 1 | 1 |
| Biologicals | 0 | 0 | 6 | 2 |
| 5-aminosalicylates | 0 | 0 | 1 | 3 |
| Combination | 0 | 0 | 8 | 2 |

PY: Pack year. CD: Crohn's disease. SD: standard deviation. IQR: interquartile range. *: $p < 0,05$.

included as actively smoking. Ileal biopsies of 19 current smokers and 18 never smokers with CD, and 18 current smokers and 20 never smokers without CD were collected. No selection was made based on gender. In five controls and five patients, ileal and colonic biopsies were taken from the same patient. The characteristics of the ileal biopsy donors are listed in Table 1. Colonic biopsies of 20 never smokers and 20 current smokers with colonic CD, and 20 healthy current smokers and 20 healthy never smokers were sampled. The characteristics of the colonic biopsy donors are listed in Table 2. A current smoker was defined as a subject consuming at least 10 cigarettes daily. A never smoker has never smoked in his/her entire life. The CD patients were diagnosed based on the clinical, endoscopic and histological features of the

**Table 2. Characteristics of colonic biopsy donors.**

| | *Control Colon* | | *Colonic CD* | |
| --- | --- | --- | --- | --- |
| | **Never smokers** | **Current smokers** | **Never smokers** | **Current smokers** |
| Number of subjects | 20 | 20 | 20 | 20 |
| Pack Year (mean, IQR) | - | 17,1 (10–40) | - | 13 (7,75–34) |
| Age (median, IQR) in years | 55,5 (31–74) | 50 (42,5–76) | 34,5 (28–60) | 39 (30–51) |
| Age at diagnosis (median, IQR) in years | - | - | 26,5 (20–56) | 31 (22–46) |
| Gender (Female/Male) | 11/9 | 12/8 | 11/9 | 7/13 |
| C-reactive protein (mean ± SD, IQR) in mg/dl | 1,07 ± 2,24 (0,1–9,3) | 2,3 ± 4,41 (0,1–14,8) | 4,49 ± 12,55 (0,5–52,5) | 11,77 ± 12,01 (1,35–37,1) |
| **Medication** | | | | |
| No Medication | 20 | 20 | 3 | 5 |
| Immunosuppressives | 0 | 0 | 2 | 7 |
| Corticosteroids | 0 | 0 | 1 | 1 |
| Biologicals | 0 | 0 | 4 | 3 |
| 5-aminosalicylates | 0 | 0 | 3 | 2 |
| Combination | 0 | 0 | 6 | 2 |

PY: Pack year. CD: Crohn's disease. SD: standard deviation. IQR: interquartile range. *: $p < 0,05$.

Lennard-Jones' criteria [28]. Control biopsies were obtained from patients who underwent colonoscopy for follow-up of polyp detection or colon carcinoma screening. The controls do not suffer from any inflammatory diseases. The samples were stored in RNAlater at -80˚C for qPCR analysis or formalin-fixed, paraffin-embedded and evaluated for histologic inflammation. TRPV1 and IL-8 expression was investigated by immunohistochemistry and qPCR. This study was approved by the local Ethical Committee of University Hospital Ghent (EC UZG 2010/116) and involved Caucasian participants. The above described methods were carried out in accordance with the approved guidelines. Written informed consent was obtained from all participating subjects.

## Animals

Male C57BL/6 mice were purchased from Charles River Laboratories (Beerse, Belgium). All mice were 6–8 weeks old at the start of the cigarette smoke exposure. The mice were housed in groups of 5 mice per cage, containing untreated wood shavings and a plastic house for environmental enrichment. The animal room was controlled and maintained at a temperature of 22˚C, humidity of 50% and a 12h/12h light/dark cycle. All mice had free access to water and were offered a standard chow diet *ad libitum* (Carfil Quality, Turnhout, Belgium). Each group consists of 10 mice. Two groups of 10 mice were used in the six-month cigarette smoking experiment and four groups of 10 mice were used in the TNBS experiment. In total, 60 mice were sacrificed. The Ethical Committee for animal experimentation of the faculty of Medicine and Health Sciences (Ghent, Belgium) approved all experiments (ECD 25/11). The above described methods were carried out in accordance with the approved guidelines of the Ethical Committee.

## Cigarette smoke exposure

Mice were exposed whole body to mainstream cigarette smoke, as described previously [29]. Briefly, groups of 10 mice were exposed to the tobacco smoke of five cigarettes (Reference Cigarette 3R4F without filter; University of Kentucky, Lexington, KY, USA) four times a day with 30 min. smoke-free intervals, five days per week for 4 or 24 weeks. An optimal smoke:air ratio of 1:6 was obtained. The control groups were exposed to air. Carboxyhaemoglobin in serum of CS-exposed mice reached a non-toxic level of 8.7 ± 0,31% (compared with 0.65 ±0,25% in air-exposed mice), which is similar to carboxyhaemoglobin blood concentrations of human smokers [30].

## Experimental colitis

After four weeks of smoke exposure, colitis was induced one day after cessation of exposure by intrarectal administration of 100 µl of 2.5% (w/v) TNBS (5% picrylsulfonic acid solution; Sigma Aldrich, Zwijndrecht, the Netherlands) diluted 1:1 in absolute ethanol, as described previously [21]. Mice were fasted one day before TNBS administration. Anesthetized mice were given either a 100 µl TNBS or sham (PBS diluted 1:1 in absolute ethanol) enema. To minimize excretion of TNBS solution, animals were inverted for 60 seconds following completion of the enema. Body weight was monitored throughout the study. At day two post-colitis induction, mice were sacrificed. The mice were euthanized with an overdose of pentobarbital (Sanofi-Ceva, Paris, France). The distal colon was excised and its length was measured, after which colonic samples were fixed in 4% (w/v) paraformaldehyde, stored in RNAlater (Qiagen, Hilden, Germany) or snap-frozen.

## Cell culture

The human colorectal adenocarcinoma cell lines HT-29 (HTB-38™) [31], Caco-2 (HTB-37™) [31] and T-84 (CCL-248™) [32] were purchased from the American Type Culture Collection (ATCC, Rockville, MD). Caco-2 was maintained in Dulbecco's Modified Eagle Medium (DMEM) with 10% foetal calf serum (FCS). T84 was maintained in Dulbecco's Modified Eagle Medium/Nutrient mixture F-12 with 5% FCS. HT-29 was maintained in DMEM containing 10% FCS, 100 U/mL penicillin, 100 μg/mL streptomycin, 2 mM L-glutamine, and 0.1 mM non-essential amino acids. All cell cultures were grown at 37˚C and a humidified atmosphere of air/$CO_2$ (95:5, v/v), with three medium changes per week. All media and supplements were purchased from Life Technologies (Merelbeke, Belgium). To induce differentiation, T-84 cells were seeded on 24-well, 0.4 μm pore diameter, semipermeable inserts (Greiner Bio-One, Vilvoorde, Belgium) at a density of $10^5$ cells per well and cultured for 3 weeks. Medium was changed three times per week. After this period, the integrity of the monolayer was evaluated by measuring the TEER using a Millicell ERS-2 Volt-Ohm Meter (Merck Millipore, Billerica, MA, USA) to ensure monolayers with TEER values of 700 Ω.cm$^2$ or higher were obtained [33]. For TRPV1 staining, HT-29 and T-84 cells were plated on glass chamber slides (Fisher Scientific, Merelbeke, Belgium) at a cell density of $5 \times 10^4$/well and cultured for 24 hours, after which the cells were harvested and washed with ice-cold PBS and fixed with 4% paraformaldehyde, while being kept in the chamber slides.

## H&E staining and scoring

Paraffin-embedded 4 μm tissue sections were taken from ileal and colonic tissue samples, dewaxed and rehydrated. The sections of human and mouse were both stained with H&E. The H&E staining was performed using the Tissue-Tek Prisma/Film automated slide stainer (Sakura, Torrance, US). Only in TNBS-treated mice, the degree of colon inflammation was scored in a blinded manner and independently by two pathologists according to a scoring scheme (Table 3) adapted from Van der Sluis et al., 2006 [34]. The histologic scoring was performed on the two most distal colonic sections, 5 mm apart from each other, and on a proximal section at approximately 5 cm from the rectum.

## Immunohistochemistry/fluorescence for TRPV1

Paraffin-embedded sections of human and mouse ileum and colon were dewaxed, after which treatment with Tris/EDTA (pH = 9) was performed for antigen retrieval. Endogenous peroxidase activity was blocked with 0.3% $H_2O_2$, followed by blocking of non-specific binding sites with 1% BSA in PBS. Subsequently, slides were incubated with the primary antibody

**Table 3. Histologic score to quantify the degree of gastrointestinal inflammation.**

| | Score | | | | |
|---|---|---|---|---|---|
| | **0** | **1** | **2** | **3** | **4** |
| Goblet cells | - | | | | |
| Mucosa thickening | - | | | | |
| Inflammatory cells | - | | | | |
| Submucosa cell infiltration | - | - | | | |
| Destruction of architecture | - | - | | | |
| Ulcers (epithelial cell surface) | 0% | 0–25% | 25%-50% | 50%-75% | 75%-100% |
| Crypt abscesses | 0 | 1–3 | 4–6 | 7–9 | >10 |

Scoring scheme adapted from Van der Sluis et al., 2006.

(polyclonal rabbit anti-TRPV1, ab31895 for mouse and ab63083 for human, Abcam) or rabbit IgG isotype control at room temperature for one hour and with the biotinylated goat anti-rabbit secondary antibody (DAKO Agilent, Santa Clara, US) for 30 min at room temperature. Thereafter, HRP-conjugated streptavidin (DAKO Agilent, Santa Clara, US) was applied for 30 min. DAB was used as an enzyme substrate before counterstaining with haematoxilin. The trigeminal ganglia of a wild-type mouse were used as a positive control. TRPV1 protein expression was quantified in an area containing 10 aligning longitudinal villi using the AxioVision software (Carl Zeiss, Zaventem, Belgium). For normalization, the measured area staining positive for TRPV1 was divided by the total area containing the 10 longitudinal villi.

TRPV1 staining of HT-29 and T-84 cells was performed by blocking paraformaldehyde-fixed cells with 1% BSA/PBS and incubating with anti-human TRPV1 (ab63083, Abcam, Cambridge, UK; 1:100) overnight at 4˚C. Detection was performed with an AlexaFluor594-conjugated secondary antibody (Life Technologies) and DAPI (1:5000) as a nuclear stain. Stained cells were viewed with a fluorescence microscope (Carl Zeiss, Zaventem, Belgium).

## Quantitative real-time PCR

RNA from ileum, proximal and distal colon of mice and human ileal and colonic biopsies was extracted using the Qiagen miRNeasy Mini Kit (Qiagen, Hilden, Germany). Subsequently, cDNA was synthesized by reverse transcription using the iScript™ cDNA Synthesis kit (Bio-Rad Laboratories, Nazareth, Belgium) following the manufacturer's instructions. Expression of mouse target genes *Kc*, *Cxcl2*, *Il-1β*, *Trpv1 and* human target genes *TRPV1* and *IL-8*, and reference genes high mobility group 20a (*Hmg20a*), hydroxymethylbilane synthase (*Hmbs*) and glyceraldehyde-3-phosphate (*Gapdh*) (sequences are provided in Table 4), was analyzed by qRT-PCR using the SensiMix™ SYBR No-ROX Kit (Bioline, London, UK). qRT-PCR was performed on a LightCycler480 detection system (Roche, Vilvoorde, Belgium) with the following cycling conditions: 10 min incubation at 95˚C, 45 cycles of 95˚C for 10 seconds and 60˚C for 1 min. Melting curve analysis confirmed primer specificity. The PCR efficiency of each primer pair was calculated using a standard curve from reference cDNA. The amplification efficiency was determined using the formula 10−1/SLOPE—1.

## Protein extraction, Luminex assay and ELISA

25–30 μg of snap-frozen mouse distal colon samples were suspended in a 10 times volume of 0.9% NaCl solution with protease inhibitor Complete Roche (Roche, Vilvoorde, Belgium) and

**Table 4. Primer sequences qRT-PCR.**

| Gene Symbol | Accession Number | Forward Primer (5'-3') | Reverse primer (3'-5') | Effic | $R^2$ |
|---|---|---|---|---|---|
| **Mouse** | | | | | |
| Hmbs | NM_001110251 | AAGGGCTTTTCTGAGGCACC | AGTTGCCCATCTTTCATCACTG | 99 | 0,99 |
| Hmg20a | NM_025812 | AGTGGAGAAATACCAACAGTGGA | AAGTTTTCGTTGTCTTGGGGAT | 98 | 0,9979 |
| Cxcl1/Kc | NM_008176 | ACCGAAGTCATAGCCACACTC | TCTCCGTTACTTGGGGACAC | 95 | 0,9995 |
| Cxcl2 | NM_009140 | GCGCCCAGACAGAAGTCATAG | AGCCTTGCCTTTGTTCAGTATC | 89,2 | 0,99 |
| Il-1β | NM_000576 | CACGATGCACCTGTACGATCA | GTTGCTCCATATCCTGTCCCT | 97 | 0,9987 |
| Trpv1 | NM_001001445 | CCGGCTTTTTGGGAAGGGT | GAGACAGGTAGGTCCATCCAC | 103 | 0,9596 |
| **Human** | | | | | |
| Gapdh | NM_002046 | TGCACCACCAACTGCTTAGC | GGCATGGACTGTGGTCATGAG | 91 | 0,9936 |
| Hmbs | NM_000190 | GGCAATGCGGCTGCAA | GGGTACCCACGCGAATCAC | 101 | 0,998 |
| Trpv1 | NM_080706 | CTGCCCGACCATCACAGTC | CTGCGATCATAGAGCCTGAGG | 93 | 0,9177 |
| Il-8 | NM_000584 | TGTTCCACTGTGCCTTGGTTTC | TGTGAGGTAAGATGGTGGCTAATAC | 98 | 0,9925 |

pulverized by means of a mechanical TissueLyser LT (Qiagen, Hilden, Germany) at 50 oscillations per second for 2 minutes in pre-chilled eppendorfs. The tissue homogenates were centrifuged at 12000 g for 5 minutes at 4°C and the supernatants were stored at −20°C until further analysis.

Protein levels of mouse CXCL1/KC, CXCL2 and IL-1β were measured by means of the Luminex xMAP Technology Reader using the Magnetic Luminex Screening Assay (R&D Systems, Abingdon, UK) on a Bio-Plex™ 200 Array Reader (BioRad, Nazareth, Belgium) according to the manufacturer's instructions.

The IL-8 protein content of human cell culture supernatant was measured by ELISA according to the manufacturer's instructions (human CXCL8/IL-8 Duoset ELISA, R&D Systems, Abingdon, UK).

The CRP level in human patient blood was determined by ELISA in the Clinical Biology Lab of the Ghent University Hospital.

### Statistical analysis

Gene expression levels depicted in bar graphs were expressed as mean ± standard error of the mean (= SEM) and error bars depict the SEM. Statistical analysis was performed using ANOVA following post-hoc Tukey tests or Student's t-test. Gene expression levels depicted in boxplots were expressed as median ± error. Statistical analysis was performed using a general linear model with smoking, IBD status and their interaction as independent variables, and either protein or gene expression level as dependent variable using R version 3.60. Multiple comparison to determine statistically significant pairs was performed using the *multicomp* package (S1 File). A p-value of less than 0.05 was considered significant. We applied the Intraclass Correlation Coefficient in order to determine correlation between histologic scores assessed by two independent pathologists.

## Results

### Smoking affects IL-8 and TRPV1 expression in the human gut

To address the effect of smoking on IL-8 and TRPV1 expression in the human healthy and inflamed gut, we analyzed ileal and colonic biopsies of 155 subjects (Tables 1 and 2). Pack years were comparable among current smokers and age of diagnosis was equal among CD patients. The mean age of the subjects was comparable within the CD and control groups, with a lower median age for CD patients compared to the healthy controls. As each group contains a wide range of ages (shown by IQR in Tables 1 and 2), age was taken into account as a confounding factor in the statistical analysis.

C-reactive protein (CRP) levels were not significantly different among ileal CD patients and their controls. In the ileal CD patient groups, never smokers mostly receive biological or combination therapy, while current smokers mostly receive immunosuppressives. In agreement with literature, *IL-8* mRNA was strongly induced in ileal biopsies of CD patients compared to healthy controls (both never and active smokers) (Fig 1A). *IL-8* mRNA is marginally induced in ileal biopsies of healthy active smokers compared to healthy never smokers. In CD, *IL-8* levels are similar in ileal biopsies of never and active smokers (Fig 1A). In the ileum, TRPV1 mRNA is elevated in never smoking healthy controls compared to never smoking CD patients (Fig 1C). Expression of TRPV1 mRNA and protein remained unchanged in the ileum (Fig 1C and 1E). A TRPV1-stained ileal biopsy of a currently smoking CD patient is shown in Fig 1G.

In currently smoking colonic CD patients, CRP levels were significantly increased compared to never smoking CD patients. In agreement with literature, *IL-8* mRNA was strongly induced in colonic biopsies of CD patients compared to healthy controls (Fig 1B). In CD,

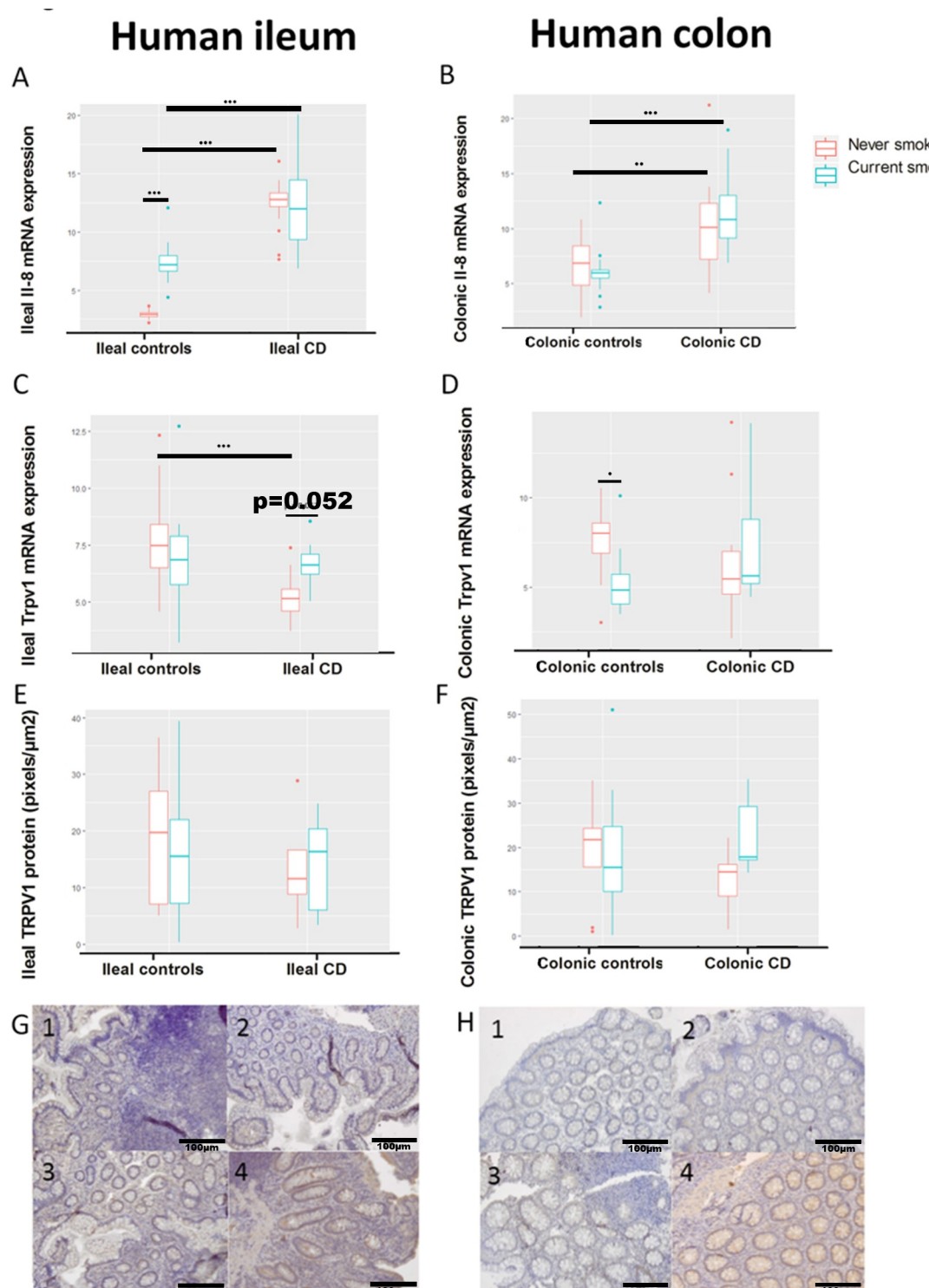

**Fig 1. mRNA expression of Il-8 and mRNA and protein expression of TRPV1 in the ileum and colon of never and currently smoking healthy subjects and Crohn's Disease (CD) patients.** mRNA values are relative to the expression of two reference genes (glyceraldehyde-3-phosphate dehydrogenase (GAPDH) and hydroxymethylbilane synthase (HMBS)). TRPV1 protein values were measured via immunohistochemical staining. (A) Expression of *Il-8* mRNA in the ileum significantly increases in currently smoking controls (p = 0.0143) and in both never and currently smoking CD patients compared to their respective controls (both p < 0.0001). (B) Expression of *Il-8* mRNA in the colon significantly increases in currently and never smoking CD patients compared to their respective controls. (C) Expression of *Trpv1* mRNA in the ileum increased in never smoking CD patients

compared to their controls (p = 0.052) and decreased in never smoking CD patients compared to never smoking healthy controls (p < 0.001). (D) Expression of *Trpv1* mRNA in the colon decreased in active smoking healthy subjects compared to never smoking healthy subjects. (E) Expression of TRPV1 protein remained unchanged in the ileum. (F) Expression of TRPV1 protein in the colon remained unchanged. (G) Ileal biopsy of a never smoking control (1), a currently smoking control (2), a never smoking CD patient (3) and a currently smoking CD patient (4) stained for TRPV1. (H) Colonic biopsy of a never smoking control (1), a currently smoking control (2), a never smoking CD patient (3) and a currently smoking CD patient (4) stained for TRPV1. Data are represented as median±error. *: p < 0.05; **: p < 0.01; ***: p < 0.001. Potential confounding factors (age, gender) were taken into account for statistical analysis.

smoking does not show a significant increase of *IL-8* mRNA levels in colonic biopsies of active smoking CD patients compared to never smoking CD patients (Fig 1B). Expression of TRPV1 mRNA and protein remained unchanged in the colon (Fig 1D and 1F). A TRPV1-stained colonic biopsy of a currently smoking CD patient is shown in Fig 1H (S2 File).

## Chronic cigarette smoke exposure induces cytokine/chemokine expression and TRPV1 in the murine gut

To further investigate the influence of cigarette smoke (CS) on intestinal cytokines/chemokines and TRPV1 expression in the gut, we exposed C57/Bl6 mice to CS during 24 weeks as described previously [35]. The expression of the mouse IL-8 homologues chemokine (C-X-C motif) ligand 2 (CXCL2) and keratinocyte chemoattractant (KC) was evaluated. In the CS-exposed mice, *Cxcl2* mRNA was increased only in the ileum and *Kc* mRNA was elevated only in the distal colon (Fig 2A and 2B). Considering the increased interest in TRP channels and their potential role in pro-inflammatory responses in the gut [36], we assessed the expression of TRPV1 in the gut. Real-time qPCR on total mRNA isolated from mouse whole intestinal tissue from three specific gut regions (ileum, proximal and distal colon) demonstrated detectable levels of mouse *Trpv1* transcripts in all regions. In ileum, *Trpv1* mRNA expression was significantly elevated by CS exposure (Fig 2C). Assessing protein expression of TRPV1 by immunohistochemistry in ileal and colonic sections revealed that TRPV1 expression increased in gut epithelial cells after chronic CS exposure and was present on the surface of the gut epithelium (Fig 2F–2I), as quantified by image analysis (Fig 2D–2I) (S3 File).

## Prior cigarette smoke exposure does not affect the development of TNBS-induced colitis in mice

We evaluated the development of TNBS-induced colitis in C57BL/6 mice that were previously exposed to CS or air during 4 weeks (Fig 3A). Two days after TNBS challenge, mice developed colitis as assessed by a significant body weight loss of 20% (Fig 3B) and colon length shortening (Fig 3C) compared to sham treatment. No significant differences in weight loss were observed in TNBS-treated mice that were either air- or CS-exposed (Fig 3B). At two days post-TNBS-enema, histological inflammation and epithelial destruction was apparent in the TNBS-treated, but not in the sham-treated group, which was not significantly affected by CS exposure (Fig 3D–3F). Also, no changes in colon length due to CS exposure were observed (Fig 3C) (S4 File).

## Prior cigarette smoke exposure aggravates TNBS-induced pro-inflammatory mediator production and TRPV1 expression in the gut

In mice exposed to air, TNBS treatment induced the expression of KC, CXCL2 and IL-1β in the distal colon, both at the mRNA and protein level two days post-TNBS (Fig 4A–4F). Exposure to CS significantly aggravated the expression of TNBS-induced *Kc*, *Cxcl2* and *Il-1*β mRNA in the distal colon after two days (Fig 4A, 4C and 4E). Also at protein level, CS exposure tended to

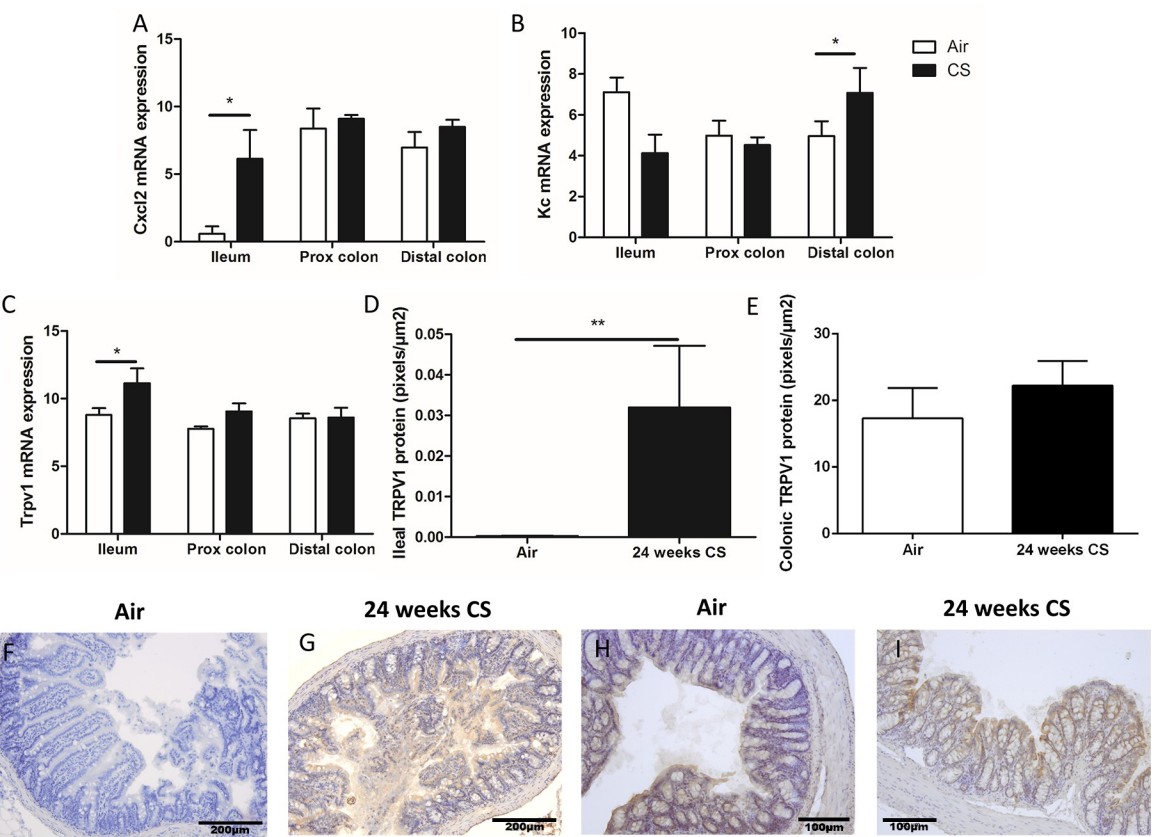

**Fig 2. Expression of cytokines/chemokines and TRP channels in non-inflamed C57/Bl6 mice chronically exposed to air or Cigarette Smoke (CS).** Relative to the expression of two reference genes (hydroxybilane synthase (HMBS) and high mobility group 20a (HMG20A)). (A) mRNA expression of *Cxcl2* significantly increases in the ileum after 24 weeks of CS exposure (p = 0.0075). (B) mRNA expression of *Kc* significantly increases in the distal colon after 24 weeks of CS exposure (p = 0.0186). (C) mRNA expression of *Trpv1* significantly increases in the ileum after 24 weeks of CS exposure (p = 0.0315). (D) Protein expression of TRPV1 significantly increases in the ileum after 24 weeks of CS exposure (p = 0.0317). (E) Protein expression of TRPV1 remains unchanged in the colon. (F) TRPV1 protein in ileum section of an air-exposed mouse. (G) TRPV1 protein in ileum section of a mouse exposed to CS during 24 weeks. (H) TRPV1 protein in colon section of an air-exposed mouse. (I) TRPV1 protein in colon section of a mouse exposed to CS during 24 weeks. ANOVA was performed. P-values lower than 0.05 were considered significant. Data are represented as mean±SEM. *: p < 0.05; **: p < 0.01; ***: p < 0.001.

induce the expression of KC and CXCL2, and significantly increased IL-1β in distal colonic tissue of TNBS-challenged mice (Fig 4B, 4D and 4F). A significant increase could not be shown for KC and CXCL2 protein, probably due to the small sample size of 10 mice/group. Notably, CS exposure during 4 weeks also modulated pro-inflammatory mediator expression in sham-treated animals, with nominal increases in KC, CXCL2 and IL-1β either at mRNA and protein level. Furthermore, we demonstrated that *Trpv1* mRNA expression is induced in the distal colon of TNBS-challenged CS-exposed mice compared to the air-exposed mice two days post-TNBS (Fig 4G), which is in line with the CS-induced aggravation of TNBS-induced CXCL2, KC and IL-1β (Fig 4A–4F). No changes in TRPV1 protein were observed in CS-exposed mice two days post-TNBS compared to sham-treated mice (Fig 4H) (S4 File).

## The TRPV1 channel is expressed by gut epithelial cells

To assess whether TRPV1 is specifically expressed by epithelial cells in the gut, as we observed in mouse ileum (Fig 2F–2I), we performed real-time qPCR on total mRNA isolated from

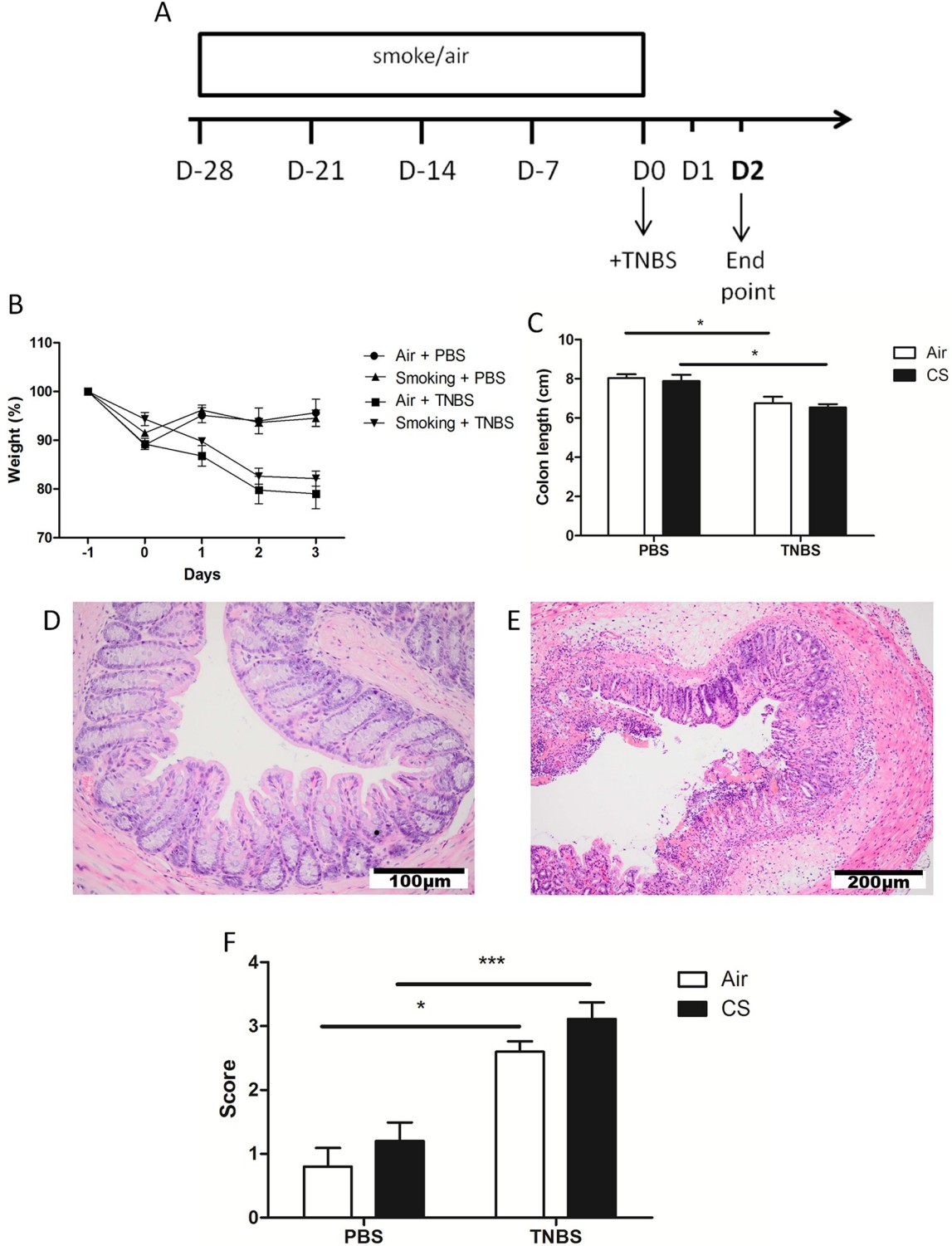

**Fig 3. Effect of CS exposure on histology and clinical parameters of TNBS-induced colitis.** (A) The applied protocol for CS exposure and TNBS-induced colitis. Mice were exposed to CS four times a day during four weeks. The day after the last CS exposure, TNBS was administered intrarectally. Mice were sacrificed at day two (D2). (B) Body weight displayed as percentage of the weight at the day of the last CS exposure (D-1). (C) Colon length (in cm) at two days post-TNBS treatment. Colons were excised from anus to caecum. Before measuring the length, the rectum (5 mm starting from the anus) was removed. (D) Distal colon of the PBS-challenged control group. (E) Distal colon of an air-exposed mouse two days post-TNBS enema. (F) Histological score of inflammation in the distal colon at two days

post-TNBS treatment. The independent investigators show a kappa of 0,287. ANOVA was performed. Data are represented as mean ±SEM. *: $p < 0.05$; **: $p < 0.01$; ***: $p < 0.001$.

whole intestinal tissue from three specific gut regions (ileum, proximal and distal colon) and the human gut epithelial cell lines HT-29, Caco-2 and T-84. We found detectable levels of TRPV1 transcripts in both human and mouse gut tissue (Figs 1, 2 and 4). *In vitro*, HT-29, Caco-2 and T-84 cells show the expression of *Trpv1* mRNA (Fig 5A). Trpv1 mRNA expression was highest in T-84 cells, especially upon differentiation. We observed the expression of TRPV1 protein in HT-29 as well as undifferentiated and differentiated T-84 (Fig 5B–5D) (S5 File).

## Discussion

In this study, we demonstrate that CS exposure affects cytokine/chemokine profiles, in line with epithelial TRPV1 expression, in the healthy and inflamed gut. We first showed that active smoking of human subjects induces IL-8 expression in the gut. In addition, a decrease in TRPV1 mRNA was found in never smoking patients suffering from Crohn's ileitis and in the colon of healthy active smokers. Second, we found that CXCL2, a mouse IL-8 homologue, and epithelial TRPV1 are simultaneously induced in the ileum of mice chronically exposed to CS. Furthermore, using a TNBS-induced gut inflammation model, we showed that, although that CS did not affect disease phenotype at a macroscopic level, it did increase CXCL2, KC and IL-1β, and TRPV1 levels in the inflamed distal colon of mice. Therefore, we suggest that CS exposure affects cytokine/chemokine profiles in healthy and inflamed gut, and concurrent CS-augmented TRPV1 may point to a correlation between IL-8 and TRPV1 increase.

A first striking finding was the induction of IL-8 in the gut of human active smokers. IL-8, and its murine homologues CXCL2 and KC, are key inflammatory mediators in acute mucosal inflammation, being involved in chemoattraction of neutrophils to the site of inflammation [12, 37]. In our study, IL-8 was increased in human active smokers depending on the gut location. In the ileum, IL-8 was augmented in actively smoking controls, while in the colon, IL-8 was elevated in actively smoking CD patients. This shows that CS boosts IL-8 expression in the human gut, which might promote neutrophil recruitment. It has previously been reported that IL-8 is increased in gut mucosal biopsies of smokers [38], which we now confirmed on a dataset of 155 patients. A study by Mortaz et al. states that cigarette smoke extract elicits an induction of IL-8 in plasmacytoid dendritic cells, resulting in IL-8-induced neutrophil recruitment [39]. However, also the epithelium can be a source of IL-8 production [40]. Using a murine model of chronic CS exposure (24 weeks), we are the first to demonstrate that both the functional IL-8 homologues CXCL2 and KC were increased in the ileum and colon respectively of mice chronically exposed to CS. Similar increases in CXCL2 and KC were observed in the distal colon in the 4-week-model of CS exposure. Taken together, these data indicate that CS could promote the attraction of neutrophils towards the ileal and colonic mucosa via the production of chemoattractants.

Another important finding was that CS aggravates the TNBS-induced cytokine and chemokine profile in the distal colon two days after initiation of inflammation. The cytokines and chemokines KC, CXCL2 and IL-1β were already increased due to TNBS administration. When combined with CS exposure, mRNA and protein levels of KC, CXCL2 and IL-1β were further augmented in the distal colon of TNBS-challenged mice. This suggests that CS is an additive factor for cytokine/chemokine production and boosts neutrophil and macrophage activity, resulting in increased cytokine/chemokine production. IL-1β, produced by macrophages through activation of the inflammasome and thereby promoting inflammation, is also

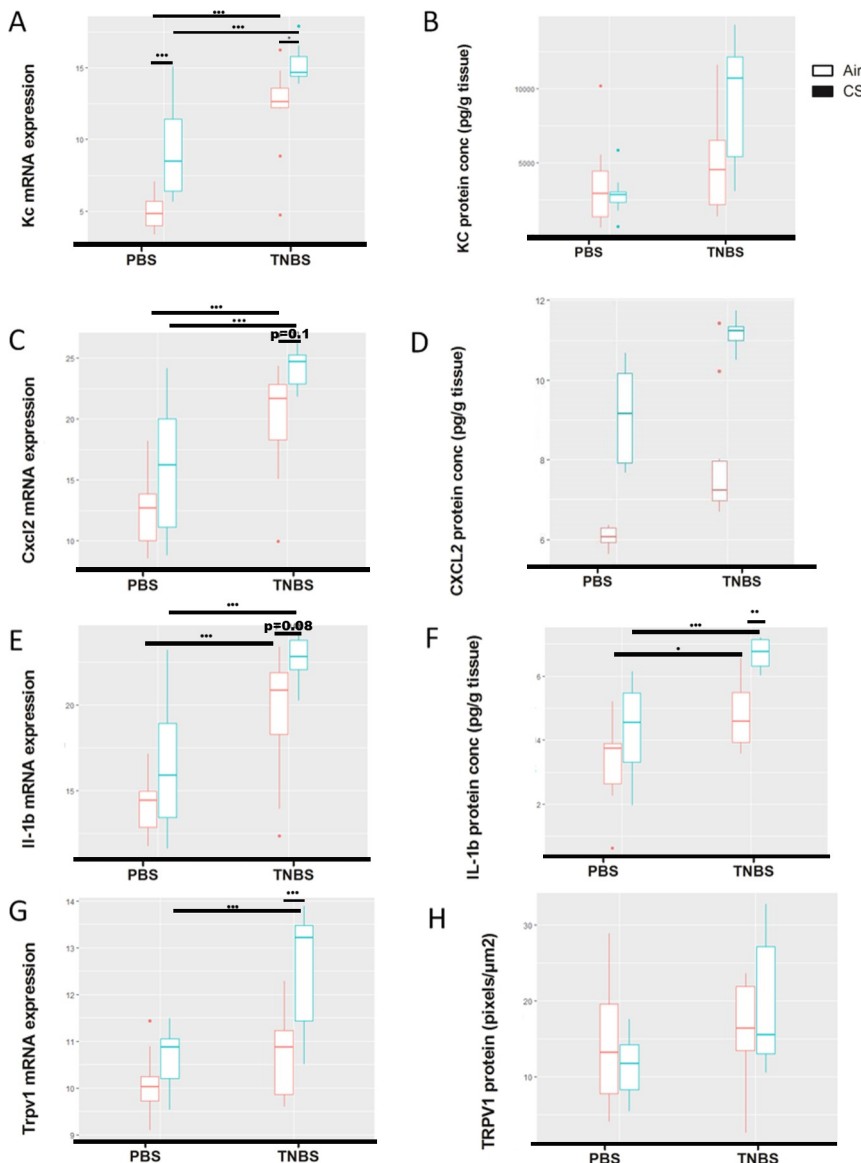

**Fig 4. Inflammatory gene expression levels in TNBS-challenged mice.** mRNA expression of cytokines in the distal colon of 10 mice/group was analyzed using real-time qPCR at two days post-TNBS-treatment. Expression levels were normalized to the reference genes high mobility group 20a (HMG20a) and hydroxymethylbilane synthase (HMBS). Protein expression of cytokines (CXCL2, KC and IL-1β) in the distal colon of 10 mice/group was analyzed using the Luminex technology and values are expressed in pg/ml tissue homogenate. Protein expression of TRPV1 in the distal colon of 10 mice/group was analyzed by microscopy and values are expressed in pixels/μm². (A) mRNA expression of *Cxcl1/Kc* increases in response to TNBS administration and after smoke exposure in both PBS- and TNBS-treated mice. (B) Protein expression of CXCL1/KC tends to increase in response to TNBS administration in CS-exposed mice and after smoke exposure in TNBS-treated mice. (C) mRNA expression of *Cxcl2* increases in response to TNBS administration and after smoke exposure in both PBS- and TNBS-treated mice. (D) Protein expression of CXCL2 tends to increase in response to TNBS administration and after smoke exposure in both PBS- and TNBS-treated mice. Data were log-transformed. (E) mRNA expression of *Il-1β* increases in response to TNBS administration and tends to increase after smoke exposure only in TNBS-treated mice. (F) Protein expression of IL-1β increases in TNBS-challenged smoke-exposed mice and after smoke exposure in both PBS- and TNBS-treated mice. Data were log-transformed. (G) mRNA expression of *Trpv1* increases in CS-exposed TNBS-challenged mice. (H) Protein expression of TRPV1 remained unchanged in TNBS-challenged smoke-exposed mice compared to smoke-exposed controls. Linear regression analysis was performed. Data are represented as median±error. *: $p < 0.05$; **: $p < 0.01$; ***: $p < 0.001$.

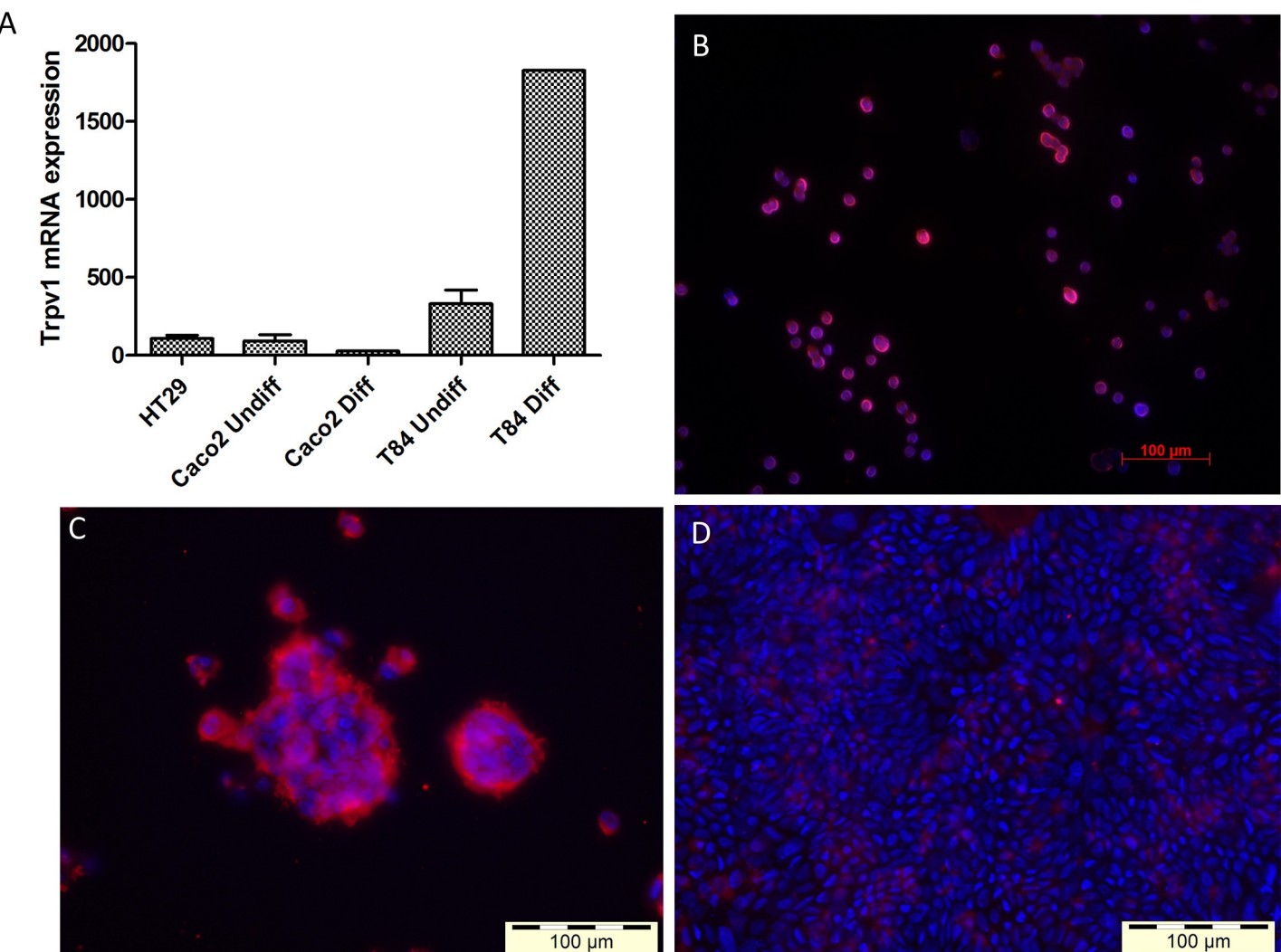

**Fig 5. mRNA and protein expression of TRPV1 in gut epithelial cell lines.** mRNA values are relative to the expression of two reference genes (glyceraldehyde-3-phosphate dehydrogenase (GAPDH) and hydroxymethylbilane synthase (HMBS)). TRPV1 protein was assessed via fluorescent staining. (A) mRNA expression level of TRPV1 in HT-29, Caco-2 and T-84. (B) Fluorescent image of HT-29 cells stained for TRPV1. (C) Fluorescent image of undifferentiated T-84 cells stained for TRPV1. (D) Fluorescent image of differentiated T-84 cells stained for TRPV1. Red: TRPV1. Blue: nuclei.

known to be raised in a TNBS-challenged colon [41, 42]. Also, it is known that neutrophils are involved in the potentiating effects of acute CS exposure on TNBS colitis in rats [43–45].

Despite the increases in cytokine/chemokine production by CS exposure, we could not show any further aggravation of histological inflammation or clinical signs such as weight loss. Hitherto, the effect of CS exposure or its components has been studied in several animal models for inflammatory bowel disease, among which TNBS-induced colitis, yielding ambiguous results [15]. Previous studies have shown a potentiating effect of CS on TNBS-induced colitis, at three days post-enema, demonstrating a promotion of neutrophil infiltration and free radical production, an upregulated expression of the α-7-nicotinic acetylcholine receptor, depletion of glutathione and overproduction of leukotriene B4 [12, 16, 18, 43]. However, in contrast to our smoke exposure model, these studies did not investigate the effect of four weeks prolonged CS exposure prior to the administration of TNBS.

In addition, we show that the TRP channel TRPV1 is expressed by the ileal and colonic epithelium, confirming the study by Kun et al. describing the presence of the channels TRPA1 and TRPV1 on colonic epithelial cells, macrophages and enteric ganglia [46]. We detected their expression in both ileum and colon of mice, human ileal and colonic biopsies and HT-29, Caco-2 and T-84 epithelial cells *in vitro*. To date, TRP channels are known as nociceptive receptors expressed by sensory neurons, for instance being involved in neurogenic inflammatory processes in the distal colon [27, 47].

Interestingly, TRPV1 expression appeared to be modulated in conditions of inflammation and CS exposure in the intestine of both humans and mice. In CD patients who have never smoked, we found a decrease in ileal and colonic *TRPV1* mRNA compared to their controls, probably due to the destruction of epithelium in the inflamed intestinal regions. It has been shown previously that TRPV1 is naturally activated in response to damage and repair [48], however mRNA levels are not sufficient to support a functional role for TRPV1. In the colon, a decrease in *TRPV1* mRNA was denoted in actively smoking human subjects compared to never smokers. These interesting human data prompted us to further analyze the effect of CS on *Trpv1* expression using experimental mouse models. We showed that chronic smoke exposure of non-inflamed mice induced ileal *Trpv1* expression. In the distal colon of TNBS-challenged mice, *Trpv1* is solely increased by four weeks prior CS exposure, suggesting a synergistic effect of CS and TNBS in inducing *Trpv1* expression.

Furthermore, a concurrent modulation of IL-8 and TRPV1 occurs in the ileum of active smokers. Although CS-induced expression of IL-8 and TRPV1 was not strictly co-regulated in the ileum and colon of human subjects, CS-induced TRPV1 expression paralleled cytokine/chemokine induction in the ileum of mouse. TRPV1 and CXCL2 were simultaneously induced in the ileum of mice chronically exposed to CS. Also, the CS-mediated aggravation of TNBS-induced CXCL2 in the distal colon of inflamed mice parallels CS-induced *Trpv1* mRNA in the distal colon of TNBS-inflamed mice. The discordance between human and mouse data may be due to the nature of human and mouse studies: mouse experiments are performed in a controlled environment, while the human population, even within one study, is very heterogeneous. Also, protein data did not fully confirm the mRNA data, which might indicate that the power of the study is a limitation. In addition, mRNA presence does not always correlate to protein expression and may depend on the half-life of the particular proteins and mechanisms such as mRNA degradation and stability, post-transcriptional modification, RNA transport and processing.

Duration of CS exposure, presence of inflammation and gut location determine whether TRPV1 is simultaneously increased with cytokine/chemokine production. It might be that CS-induced increase of IL-8 and TRPV1 is linked. However, upregulation of TRPV1 could also be a consequence of inflammation, e.g. in TNBS-induced colitis, although CS on its own only boosts cytokine production without any histological damage. Future research is needed, exposing TRPV1-/- mice and wild-type mice treated with TRPV1 inhibitors and agonists to CS, to investigate the potential link between TRPV1 and inflammatory mediator production.

Next to local cytokine induction in the human gut, the effect of smoking on the immune system is also reflected in CRP levels in the serum. Moreover, in currently smoking colonic CD patients, higher CRP levels were found. In ileal CD patients, the increase was not significant, probably due to high inter-patient variation. A larger cohort would be needed. Furthermore, our data might show a smoking-dependent shift in the treatment strategy of CD patients with ileal involvement: never smokers mostly receive biological or combination therapy, while current smokers mostly receive immunosuppressives. However, a multi-center study with a larger cohort would be needed to present significant conclusions. The current data show that CS exposure seemingly acts as a trigger of the gut immune system, serving as a

predisposing environmental factor for potential development of inflammation, which corresponds with our previous findings that chronic CS exposure in mice triggers the recruitment of immune cells to the ileal Peyer's patches [14].

In conclusion, we demonstrated, using human gut samples as well as murine models, that CS modulates pro-inflammatory cytokines/chemokines and suggest a link between inflammation and TRPV1 expression in the gut. CS-augmented expression of cytokines/chemokines and TRPV1 often occurs simultaneously, suggesting a link between both. Future research is needed to investigate whether TRPV1 might be involved in CS-increased production of cytokines and chemokines.

## Supporting information

**S1 File. Regression analysis.**
(PDF)

**S2 File. Human study.**
(XLSX)

**S3 File. Mouse study–long-term smoking.**
(XLSX)

**S4 File. Mouse study–TNBS experiment.**
(XLSX)

**S5 File. Cell line experiments.**
(XLSX)

**S1 Fig. Weight loss in healthy CS-exposed mice.**
(TIF)

## Acknowledgments

We thank Ran Rumes and Lynn Supply for the support with the animal experiments and the processing of the samples, and Eliane Castrique, Katleen de Saedeleer, Anouk Goethals, Ann Neesen, Indra de Borle, Evelyn Spruyt, Greet Barbier and Lien Coulembier for the excellent technical support with the animal experiments. We are very grateful to Gabrielle Holtappels for the technical assistance with the Luminex assays. We thank Elien Glorieus, Pieter Hindryckx and Koen Gorleer for contributing to the collection of the human samples.

## Author Contributions

**Conceptualization:** Liesbeth Allais, Stephanie Verschuere, Tania Maes, Harald Peeters, Martine De Vos, Guy G. Brusselle, Ken R. Bracke, Claude A. Cuvelier, Debby Laukens.

**Data curation:** Liesbeth Allais, Claude A. Cuvelier, Debby Laukens.

**Formal analysis:** Liesbeth Allais, Rebecca De Smet, Sarah Devriese, Gerard Bryan Gonzales, Guy G. Brusselle, Claude A. Cuvelier, Debby Laukens.

**Funding acquisition:** Liesbeth Allais, Stephanie Verschuere, Martine De Vos, Guy G. Brusselle, Ken R. Bracke, Claude A. Cuvelier, Debby Laukens.

**Investigation:** Liesbeth Allais, Sarah Devriese, Ken R. Bracke, Debby Laukens.

**Methodology:** Liesbeth Allais, Tania Maes, Sarah Devriese, Gerard Bryan Gonzales, Harald Peeters, Koen Van Crombruggen, Claus Bachert, Martine De Vos, Guy G. Brusselle, Ken R. Bracke, Claude A. Cuvelier, Debby Laukens.

**Project administration:** Liesbeth Allais, Rebecca De Smet, Martine De Vos, Claude A. Cuvelier, Debby Laukens.

**Resources:** Liesbeth Allais, Guy G. Brusselle, Ken R. Bracke, Claude A. Cuvelier, Debby Laukens.

**Software:** Liesbeth Allais, Rebecca De Smet, Martine De Vos, Ken R. Bracke, Claude A. Cuvelier, Debby Laukens.

**Supervision:** Stephanie Verschuere, Tania Maes, Martine De Vos, Guy G. Brusselle, Ken R. Bracke, Claude A. Cuvelier, Debby Laukens.

**Validation:** Liesbeth Allais, Ken R. Bracke, Claude A. Cuvelier, Debby Laukens.

**Visualization:** Liesbeth Allais, Claude A. Cuvelier, Debby Laukens.

**Writing – original draft:** Liesbeth Allais, Martine De Vos, Guy G. Brusselle, Ken R. Bracke, Claude A. Cuvelier, Debby Laukens.

**Writing – review & editing:** Liesbeth Allais, Stephanie Verschuere, Tania Maes, Rebecca De Smet, Gerard Bryan Gonzales, Koen Van Crombruggen, Claus Bachert, Martine De Vos, Guy G. Brusselle, Ken R. Bracke, Claude A. Cuvelier, Debby Laukens.

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
