## [Decision Letter · Decision Letter 0]

28 May 2020

PONE-D-20-04633

Translational research into the effects of cigarette smoke on inflammatory mediators and epithelial TRPV1 in Crohn’s disease

PLOS ONE

Dear Dr. Allais,

Thank you for submitting your manuscript to PLOS ONE. After careful consideration, we feel that it has merit but does not fully meet PLOS ONE’s publication criteria as it currently stands. Therefore, we invite you to submit a revised version of the manuscript that addresses the points raised during the review process.

We look forward to receiving your revised manuscript.

Kind regards,

Mathilde Body-Malapel

Academic Editor

PLOS ONE

Reviewers' comments:

Reviewer's Responses to Questions

**Comments to the Author**

1. Is the manuscript technically sound, and do the data support the conclusions?

Reviewer #1: Partly

Reviewer #2: No

2. Has the statistical analysis been performed appropriately and rigorously? 

Reviewer #1: I Don't Know

Reviewer #2: No

3. Have the authors made all data underlying the findings in their manuscript fully available?

Reviewer #1: No

Reviewer #2: No

4. Is the manuscript presented in an intelligible fashion and written in standard English?

Reviewer #1: No

Reviewer #2: Yes

5. Review Comments to the Author

Reviewer #1: In this work, Allais et. al analyzed the expression of TRPV1 and cytokines/chemokines in intestinal mucosa after exposure to cigarette smoke, in subjects (patients and mice) with or without IBD. The intention to integrate results obtained in humans and IBD mice model is highlighted. However, the manuscript is unclear and various details are missing to validate the results and to integrate both models. Indeed, more analyzes are needed to validate the conclusions.

Major comments:

Increased TRPV1 does not necessarily imply increased cytokine expression. TRPV1 is naturally activated in response to damage and repair (PMID: 25083990), so it may be just parallel processes. An experiment in vivo with TRPV1 inhibitor is required (or at least in vitro or ex vivo - e.g. explants with CSC) to delve into this association.

How would the effects of CS on TRPV1 expression be explained? Agonists such as leukotriene B4 need to be evaluated. There is evidence that nicotine reduces LTB4, and therefore shows beneficial effects in similar models (PMID: 26881175). Authors should discuss these differences.

TNBS-induced Colitis model does not recapitulate Crohn’s disease in terms of aetiopathogenesis. Indeed, it represents a model for acute Th1 colonic inflammation, but cigarette smoking is mostly associated with ileal inflammation. As shown in figure 1, CS was only associated with IL-8 in ileal biopsies. Considering that, authors should have used a murine model of ileal or ileo-colonic CD. Even though authors demonstrated a colonic effect of CS in TNBS-mice, this must be discussed (see PMID: 29441064).

The manuscript is unclear:

1) In most parts of the results description (Abstract and Results) it is not clear which groups were compared. E.g. L52-53 “In the ileum, TRPV1 mRNA levels were decreased in never smoking CD patients” in comparison with?. Authors should compare patients over healthy subjects, or treated over controls. It is confusing to say that not smoking reduces TRPV1 mRNA. E.g. L 289-291: “In CD, IL-8 levels are similar in ileal biopsies (Fig 1A)” (among active and never smokers?). In the ileum, TRPV1 mRNA is elevated in never-smoking healthy controls compared to never-smoking CD patients (Fig. 1C). Remember, treated group over control; or in this case: patients over healthy subjects.

2) Statistical analyzes are not well described. In the methodology authors mentioned linear models and multicomp package, but in the figures means + SEM are indicated (those boxplots must show Median + error). Authors should specify control variables (gender, age), not only in methodology, but also in figure legends (none specify the method). On the other hand, the variables and the statistical methods used to analyze the results of mice are not specified. It is suggested to use Pfaffl quantification to correct gene expression by qPCR efficiency.

3) The authors should try to better integrate the description and the results of both models (human and mice). Since the most conclusive results occurred in mice, it would be preferable to start by describing them: CS increases chemokines and TRPV1 in mice. Then, describe whether this effect alters the development of colitis: inflammation and TRPV1 expression (authors should confirm the TRPV1-inflammation relationship as mentioned above). And then continue to assess whether this association is also observed in patients. This would allow to discuss the difference between the effect of CS in ileum and colon (see PMID: 29441064) and in mice vs human: authors must discuss differences between whole body CS exposure vs cigarette smoking (e.g. absence of direct swallowing of particulate matter in murine model).

Minor comments

1. Biopsies histological damage is not indicated in results. This should be a cofactor for the analyzes to give an idea of the importance of tissue damage on the results.

2. Fig 1G: All biopsies should have the same orientation. Number 4 is almost longitudinal, so naturally the mucus will retain more secondary antibody. In addition, the description in the text must be corrected.

3. GAPDH is mentioned in results as reference gene, but not in Methods. Please clarify. On the other hand, where are the results of the other cytokines mentioned in methods?

4. In Fig 2B, CS-treatment reduced Kc mRNA in ileum, more than the increase in colon, however it was not significant. Please clarify statistical analysis.

5. IHC photos with anti-TRPV1 are not convincing. Fig 2G binding is present in the mucus suggesting some not specific binding. The control (Fig 2F) es not completely useful, because it seems to represent another portion of small intestine than 2G. The authors should include also a colonic photo of the air-control. Scale bars should be added. Analyzes should be done from at least three different sections per mouse.

6. Were clinical signs of colitis evaluated? Tenesmus, blood or mucus in stools, etc..

7. Authors should include the weight curves of mice during CS-treatment in supplementary material. CS-treatment is quite stressful, and together with the anorectic effects of tobacco, the resultant reduced ingestion could affect the basal state of the intestine. In fact, weight could be included as a covariate in case of using a linear model.

8. The results observed in cell lines are a good control but do not contribute to the conclusion. They should be included in supplementary material.

Reviewer #2: In this study, Allais et al. investigated the effect of CS exposure in CD patients and mouse model. They showed the CS induced the chemokines for neutrophil and TRPV1 in gut from CD patients and TNBS-colitis model. However, the implication and function of TRPV1 upregulation by CS exposure is not well presented. In addition, several concerns were raised.

Major comments;

Authors showed chemokines for neutrophil migration were upregulated by CS in patients and CD mice model. As pointed out by reviewer 1, assessment of neutrophils infiltration and activity were important for understanding the effect of CS. Therefore, authors should evaluate the neutrophil number in ileum and colon tissues from CD patients and TNBS-induced colitis model.

In figure 2C and 2D, TRPV1 protein level, measured by microscopic data in ileum from control mice was far apart form mRNA level. It seems measurement by pixels in microscopy was not enough to evaluate the expression level. Therefore, authors should confirm the increase of TRPV1 protein in ileum and colon in mice (Fig 2, 4）as well as patient (Fig 1) by ELISA or western blotting.

In figure 5, authors measured the TRPV1 expression in gut epithelial cell lines. Dose stimulation with CS extract induce the TRPV1 expression in these cell line?

Minor comments;

P9, line 165, the detail information of mice exposed whole body to mainstream CS is necessary.

How about gene expressions of other TRPVs, TRPV 2-6? It should be mentioned.

6. PLOS authors have the option to publish the peer review history of their article (what does this mean?). If published, this will include your full peer review and any attached files.

Reviewer #1: Yes: Loni Berkowitz

Reviewer #2: No

---

## [Author Response · Author response to Decision Letter 0]

7 Jul 2020

Dear Dr. Body-Malapel,

We are grateful for the opportunity to submit a revised version of our research article entitled “Translational research into the effects of cigarette smoke on inflammatory mediators and epithelial TRPV1 in Crohn’s disease”. We thank the reviewers for their valuable time and useful contribution in the peer review of our work and hereby provide you with a detailed response to the comments raised.

The current manuscript has been approved by all authors and all prevailing local, national and international regulations and conventions, and normal scientific ethical practices have been respected. No author has an ethical or financial conflict of interest. We affirm that this manuscript is original and has not been published previously nor that it is under consideration for publication in another journal. None of the manuscript’s contents has been previously published except in abstract form. We give consent for publication in PloS One, if the manuscript is accepted.

Sincerely yours,

Dr. Liesbeth Allais, on behalf of the authors.

We have adjusted the manuscript to PLOS ONE's style requirements.

2. We note that you have included the phrase “data not shown” in your manuscript. Unfortunately, this does not meet our data sharing requirements. PLOS does not permit references to inaccessible data. We require that authors provide all relevant data within the paper, Supporting Information files, or in an acceptable, public repository. Please add a citation to support this phrase or upload the data that corresponds with these findings to a stable repository (such as Figshare or Dryad) and provide URLs, DOIs, or accession numbers that may be used to access these data. Or, if the data are not a core part of the research being presented in your study, we ask that you remove the phrase that refers to these data.

The data referred to as ‘data not shown’ do not contribute to the overall conclusions of this objective, therefore, we have removed the phrase that refers to these data. 

3. Please include captions for your Supporting Information files at the end of your manuscript, and update any in-text citations to match accordingly. 

We have included captions and in-text citations for the supporting information files at the end of the manuscript.

Reviewers' comments:

Reviewer 1:

In this work, Allais et. al analyzed the expression of TRPV1 and cytokines/chemokines in intestinal mucosa after exposure to cigarette smoke, in subjects (patients and mice) with or without IBD. The intention to integrate results obtained in humans and IBD mice model is highlighted. However, the manuscript is unclear and various details are missing to validate the results and to integrate both models. Indeed, more analyses are needed to validate the conclusions.

Major Comments:

1. Increased TRPV1 does not necessarily imply increased cytokine expression. TRPV1 is naturally activated in response to damage and repair (PMID: 25083990), so it may be just parallel processes. An experiment in vivo with TRPV1 inhibitor is required (or at least in vitro or ex vivo - e.g. explants with CSC) to delve into this association.

We agree with the reviewer’s comment that increased TRPV1 does not necessarily imply an increased cytokine expression. The use of TRPV1 inhibitors in vivo or TRPV1 knock-out mice to substantiate this link is required, but is not the scope of this study. We have clarified this in the Discussion section.

Line 465 – 468 now reads as: “It has been shown previously that TRPV1 is naturally activated in response to damage and repair, however mRNA levels are not sufficient to support a functional role for TRPV1 and requires further study”.

2. How would the effects of CS on TRPV1 expression be explained? Agonists such as leukotriene B4 need to be evaluated. There is evidence that nicotine reduces LTB4, and therefore shows beneficial effects in similar models (PMID: 26881175). Authors should discuss these differences.

To explain a potential link between CS exposure and TRPV1 expression, further experiments using both TRPV1 inhibitors and agonists, as well as TRPV1 knock-out mice would be needed. We clarify this in the Discussion section. Also, it would be interesting to study the effect of full cigarette smoke exposure instead of solely nicotine on LTB4 expression. In case of the current human dataset, available material was limited. LTB4 will be considered for future studies.

3. TNBS-induced colitis model does not recapitulate Crohn’s disease in terms of aetiopathogenesis. Indeed, it represents a model for acute Th1 colonic inflammation, but cigarette smoking is mostly associated with ileal inflammation. As shown in figure 1, CS was only associated with IL-8 in ileal biopsies. Considering that, authors should have used a murine model of ileal or ileo-colonic CD. Even though authors demonstrated a colonic effect of CS in TNBS-mice, this must be discussed (see PMID: 29441064).

It is indeed correct that cigarette smoking is mostly associated with ileal inflammation. We have investigated the effect of cigarette smoke exposure in both ileal and colonic inflammation models. We previously elaborated on the effects of cigarette smoke in the ileum: Verschuere et al., 2012 (PMID: 22198215), Allais et al., 2016 (PMID: 26033517), Allais et al., 2017 (PMID: 27388890), which were indeed cited in the review PMID: 29441064. We also published our findings using the Crohn-like ileitis TNFΔARE mouse model: Allais et al., 2015 (PMID: 26523550). In the current paper under review, we chose to highlight the effect of cigarette smoke exposure on Crohn’s colitis. Therefore, we chose the Crohn-like colitis mouse model based on TNBS instead of e.g. the UC-like colitis mouse model based on DSS. Moreover, a true ileo-colonic CD-like mouse model is not well-described to date. Perhaps a combined T cell transfer model would be useful.

4. The manuscript is unclear: 

1) In most parts of the results description (Abstract and Results) it is not clear which groups were compared. E.g. L52-53 “In the ileum, TRPV1 mRNA levels were decreased in never smoking CD patients” in comparison with?. Authors should compare patients over healthy subjects, or treated over controls. It is confusing to say that not smoking reduces TRPV1 mRNA. E.g. L 289-291: “In CD, IL-8 levels are similar in ileal biopsies (Fig 1A)” (among active and never smokers?). In the ileum, TRPV1 mRNA is elevated in never-smoking healthy controls compared to never-smoking CD patients (Fig. 1C). Remember, treated group over control; or in this case: patients over healthy subjects.

We confirm that each statement in the Results section is based on comparisons between treated and control groups or between patients and healthy subjects. We have clarified this in the text.

Line 33-35: “In the ileum, TRPV1 mRNA levels were decreased in never smoking Crohn’s disease patients compared to healthy subjects (p <0,001; n = 20/group).” and line 273-274: “In CD, IL-8 levels are similar in ileal biopsies of never and active smokers (Fig 1A).” 

2) Statistical analyses are not well described. In the methodology authors mentioned linear models and multicomp package, but in the figures means + SEM are indicated (those boxplots must show Median + error). Authors should specify control variables (gender, age), not only in methodology, but also in figure legends (none specify the method). On the other hand, the variables and the statistical methods used to analyze the results of mice are not specified. It is suggested to use Pfaffl quantification to correct gene expression by qPCR efficiency.

Indeed, in the figures showing boxplots, Fig 1 and 4, the legend should mention that the ‘Median±error’ is indicated instead of ‘Means±SEM’. Figure 1 and 4 have been adapted accordingly. Also, the control variables (gender, age) in het human dataset were added in the legend of Figure 1. Variables and statistical methods were specified in the Methods section ‘Statistical Analysis’ and in the legends of figures depicting mouse data (Fig 2, 3, 4). Line 251-256: “Gene expression levels depicted in bar graphs were expressed as mean ± standard error of the mean and error bars depict the standard error of the mean. Statistical analysis was performed using ANOVA following post-hoc Tukey tests or Student’s t-test. Gene expression levels depicted in boxplots were expressed as median ± error. Statistical analysis was performed using a general linear model with smoking, IBD status and their interaction as independent variables, and either protein or gene expression level as dependent variable using R version 3.60.”

We agree that the Pfaffl quantification to correct gene expression by qPCR efficiency is recommended for analysis. We have performed relative quantification using two stably expressed reference genes. Six reference genes were evaluated previously and those that were most stable in all treatment groups were selected. The relative expression values were calculated using a static efficiency of 2. Each primer set is evaluated for qPCR efficiency (95% - 110%) using a standard curve of reference genomic DNA. 

3) The authors should try to better integrate the description and the results of both models (human and mice). Since the most conclusive results occurred in mice, it would be preferable to start by describing them: CS increases chemokines and TRPV1 in mice. Then, describe whether this effect alters the development of colitis: inflammation and TRPV1 expression (authors should confirm the TRPV1-inflammation relationship as mentioned above). And then continue to assess whether this association is also observed in patients. This would allow to discuss the difference between the effect of CS in ileum and colon (see PMID: 29441064) and in mice vs human: authors must discuss differences between whole body CS exposure vs cigarette smoking (e.g. absence of direct swallowing of particulate matter in murine model).

By describing the human data first, we aimed to clarify from which view point we started this study. Starting with the human data, we shift to the findings in healthy long-term smoking mice over to a Crohn-like colitis model, to end with beginning findings in human cell lines. Indeed, further experiments elaborating on TRPV1 and inflammation are required to ascribe any functional relevance. Also, the patient studies should be expanded to multi-center studies using a larger patient cohort. We are aware of the limitations of the current study, which we address in the Discussion section, line 479-486: “The discordance between human and mouse data may be due to the nature of human and mouse studies: mouse experiments are performed in a controlled environment, while the human population, even within one study, is very heterogeneous. Also, protein data did not fully confirm the mRNA data, which might indicate that the power of the study is a limitation. In addition, mRNA presence does not always correlate to protein expression and may depend on the half-life of the particular proteins and mechanisms such as mRNA degradation and stability, post-transcriptional modification, RNA transport and processing.” 

We have previously discussed the potential link between smoking, inflammation and TPRV1 (Allais et al., 2017, PMID: 27388890). The mentioned study (PMID: 29441064) indeed cites our work: Verschuere et al., 2012 (PMID: 22198215), Allais et al., 2016 (PMID: 26033517), Allais et al., 2017 (PMID: 27388890).

It is indeed correct that mice do not directly swallow the particulate matter of cigarette smoke. To align smoking regiments with human habits, the cigarette smoking model was optimized to have a similar level of carboxyhaemoglobin in the serum of cigarette smoke-exposed mice as compared to smoking humans. This is mentioned in the Methods section ‘Cigarette smoke exposure’, line 152-155: “Carboxyhaemoglobin in serum of CS-exposed mice reached a non-toxic level of 8.7 ± 0,31% (compared with 0.65 ±0,25% in air-exposed mice), which is similar to carboxyhaemoglobin blood concentrations of human smokers.”

Minor comments 

1. Biopsies histological damage is not indicated in results. This should be a cofactor for the analyses to give an idea of the importance of tissue damage on the results.

We observed typical histological lesions in the intestinal biopsies of CD patients, however, these lesions were not worsened due to smoking. In the healthy controls, no lesions were observed. We have previously shown that cigarette smoke does not cause macroscopical or microscopical damage to the gut (Verschuere et al., 2011 and Allais et al., 2015). Cigarette smoking as such does not cause any tissue damage. As we know this from previous studies, we did not mention this in the Results sections on the human data and the six-months smoking experiment with healthy mice. However, we did mention histological damage in the Results section ‘Prior cigarette smoke exposure does not affect the development of TNBS-induced colitis in mice’ (Figure 3). Line 336-344: “We evaluated the development of TNBS-induced colitis in C57BL/6 mice that were previously exposed to CS or air during 4 weeks (Fig 3A). Two days after TNBS challenge, mice developed colitis as assessed by a significant body weight loss of 20% (Fig 3B) and colon length shortening (Fig 3C) compared to sham treatment. No significant differences in weight loss were observed in TNBS-treated mice that were either air- or CS-exposed (Fig 3B). At two days post-TNBS-enema, histological inflammation and epithelial destruction was apparent in the TNBS-treated, but not in the sham-treated group, which was not significantly affected by CS exposure (Fig 3D-F). Also, no changes in colon length due to CS exposure were observed (Fig 3C) (S4 Supporting Information).” 

2. Fig 1G: All biopsies should have the same orientation. Number 4 is almost longitudinal, so naturally the mucus will retain more secondary antibody. In addition, the description in the text must be corrected.

Since biopsy material from human subjects is limited and difficult to position, the orientation of the biopsies may differ. However, in case of the mouse experiments, the mice were sacrificed and the complete gut was preserved. Therefore, we were able to make tissue sections in exactly the same orientation. This should however not interfere with scoring of the sections. 

3. GAPDH is mentioned in results as reference gene, but not in Methods. Please clarify. On the other hand, where are the results of the other cytokines mentioned in methods?

We agree with the reviewer’s comment. The primer sequences for the human reference genes GAPDH and HMBS are now added to Table 4. We mentioned the primers of all genes studied, however, for the cytokines TNF-α, CCR6, CCL19, CCL20 and TRP channel TRPA1, we couldn’t observe any differences in mRNA expression. 

Figure. mRNA expression of TNF-α in ileum, proximal and distal colon of wild-type C57/Bl6 mice after 24 weeks of CS exposure. No significant differences in expression were detected. IL: ileum. PC: proximal colon. DC: distal colon. Air: no cigarette smoke exposure. CS: cigarette smoke exposure of 24 weeks.

Figure. mRNA expression of TNF-α in the colon of TNBS-challenged wild-type C57/Bl6 mice after 4 weeks of CS exposure. No significant differences in expression were detected. PBS: phosphate buffered saline. TNBS: trinitrobenzene sulphonic acid. Air: no cigarette smoke exposure. CS: cigarette smoke exposure of 4 weeks.

As we found in our previously published studies (Verschuere et al., 2011; PMID: 21537330) that cigarette smoke exposure modulates the CCL20-CCR6 pathway, we thought it would be interesting to test whether cigarette smoke exposure would modulate expression of CCR6, CCL19 and CCL20 in the TNBS colitis model. However, no significant differences could be detected.

Figure. mRNA expression of CCR6 in the colon of TNBS-challenged wild-type C57/Bl6 mice after 4 weeks of CS exposure. No significant differences in expression were detected. PBS: phosphate buffered saline. TNBS: trinitrobenzene sulphonic acid. Air: no cigarette smoke exposure. CS: cigarette smoke exposure of 4 weeks.

Figure. mRNA expression of CCL19 in the colon of TNBS-challenged wild-type C57/Bl6 mice after 4 weeks of CS exposure. No significant differences in expression were detected. PBS: phosphate buffered saline. TNBS: trinitrobenzene sulphonic acid. Air: no cigarette smoke exposure. CS: cigarette smoke exposure of 4 weeks.

Figure. mRNA expression of CCL20 in the colon of TNBS-challenged wild-type C57/Bl6 mice after 4 weeks of CS exposure. No significant differences in expression were detected. PBS: phosphate buffered saline. TNBS: trinitrobenzene sulphonic acid. Air: no cigarette smoke exposure. CS: cigarette smoke exposure of 4 weeks.

In literature, more studies discuss TRPA1 rather than TPRV1 in the gut. Therefore, we initially investigated TRPA1, but no significant differences were detected.

Figure. mRNA expression of TRPA1 in ileum, proximal and distal colon of wild-type C57/Bl6 mice after 24 weeks of CS exposure. No significant differences in expression were detected. IL: ileum. PC: proximal colon. DC: distal colon. Air: no cigarette smoke exposure. CS: cigarette smoke exposure of 24 weeks.

As these negative data do not contribute to the paper, these results were not discussed in the results and discussion. We have omitted these primer sequences from Table 4. 

4. In Fig 2B, CS-treatment reduced Kc mRNA in ileum, more than the increase in colon, however it was not significant. Please clarify statistical analysis.

Indeed, figure 2B shows a reduction in Kc mRNA due to CS treatment. However, variation within the groups (ileum of air-exposed mice compared to ileum of CS-exposed mice) was too large so we could only detect a trend with a p-value > 0,05 (7,111±0,7166 in air-exposed ileum compared to 4,126±0,9066 in CS-exposed ileum; p = 0,07). For Kc mRNA in colonic tissue, we observed much less variation between treatment groups. For statistical analysis, ANOVA (F-test) was performed as a multiple group comparison test, in order to find significant differences between two or more population means. This was followed by post-hoc Tukey for pair-wise multiple comparison testing.

5. IHC photos with anti-TRPV1 are not convincing. Fig 2G binding is present in the mucus suggesting some not specific binding. The control (Fig 2F) is not completely useful, because it seems to represent another portion of small intestine than 2G. The authors should include also a colonic photo of the air-control. Scale bars should be added. Analyses should be done from at least three different sections per mouse.

We agree that it looks like the TPRV1-specific antibody is binding to the mucus. However, we know that TRPV1 is present on the epithelial cell membrane, so it can be expected in that location. Moreover, the statistical difference between air- and CS-exposed mice is large enough that it cannot be attributed to non-specific binding, taking into account that the air-exposed mice also produce mucus. We included an image of the air-exposed mouse colon in Figure 2. Figure 2E, F, G and I show scale bars at the bottom of the picture. We examined 10 mice per treatment group and 1 section of ileum and colon for each mouse. Indeed, additional experiments need to be performed investigating TRPV1 protein expression. 

6. Were clinical signs of colitis evaluated? Tenesmus, blood or mucus in stools, etc..

We only monitored body weight during the experiment, since this is most reflective of disease activity in this model. Histological damage and colon length were assessed as an end-point. 

7. Authors should include the weight curves of mice during CS-treatment in supplementary material. CS-treatment is quite stressful, and together with the anorectic effects of tobacco, the resultant reduced ingestion could affect the basal state of the intestine. In fact, weight could be included as a covariate in case of using a linear model.

We have indeed monitored weight during CS exposure. We weighed at different time points. After two weeks, we already observed significant weight loss in CS-exposed mice, compared to air-exposed mice. We have included the weight follow-up in supplementary data: S3 Supporting information + S6 Figure. In addition, the weight follow-up during the TNBS exposure in Figure 3B is depicted as percentages.

8. The results observed in cell lines are a good control but do not contribute to the conclusion. They should be included in supplementary material.

We are aware that the cell line data do not support the final conclusion. However, as few studies have described the presence of TPRV1 on the cell membrane of gut epithelial cell lines, we absolutely wanted to share this interesting finding.

Reviewer 2:

In this study, Allais et al. investigated the effect of CS exposure in CD patients and mouse model. They showed that CS induced the chemokines for neutrophil and TRPV1 in gut from CD patients and TNBS-colitis model. However, the implication and function of TRPV1 upregulation by CS exposure is not well presented. In addition, several concerns were raised.

Major Comments:

1. Authors showed chemokines for neutrophil migration were upregulated by CS in patients and CD mice model. As pointed out by reviewer 1, assessment of neutrophil infiltration and activity is important for understanding the effect of CS. Therefore, authors should evaluate the neutrophil number in ileum and colon tissues from CD patients and TNBS-induced colitis model.

We agree that it would be interesting and important for understanding the effect of CS to evaluate the presence of neutrophils and their activation, e.g. by measurement of MPO, and we will take this into account for future experiments. We previously elaborated on the effects of cigarette smoke on immune cell populations in the Peyer’s patches in the ileum: Verschuere et al., 2012 (PMID: 221537330). It was shown that total dendritic cell count, CD4+ and CD8+ T cells and the CD11b+ dendritic cell subset was increased in response to 24 weeks of cigarette smoke exposure in wild-type mice. No difference in neutrophil count was observed.

2. In figure 2C and 2D, TRPV1 protein level measured by microscopic data in ileum from control mice was far apart from mRNA level. It seems measurement by pixels in microscopy was not enough to evaluate the expression level. Therefore, authors should confirm the increase of TRPV1 protein in ileum and colon in mice (Fig 2, 4）as well as patients (Fig 1) by ELISA or western blotting.

We agree that ELISA or western blotting is necessary to confirm measurement by pixels in microscopy. In this study, we chose for immunohistochemistry to obtain particular information on the location of the TRPV1 protein. Many studies describe TRPV1 in the neuronal system, however, few studies have elaborated on its epithelial expression. In addition, biopsy material in the human study was limited.

3. In figure 5, authors measured the TRPV1 expression in gut epithelial cell lines. Does stimulation with CS extract induce the TRPV1 expression in these cell line?

Indeed, it would be interesting to investigate TRPV1 expression in response to stimulation with CS extract. We have started human cell line experiments (Caco2, T84, HT29) and intestinal mouse organoid culture experiments with CS extract and important CS components (nicotine, acrolein, 4-hydroxy-2-nonenal, H2O2) in different concentrations. We selected components already described in literature and practically feasible to test in cell culture experiments. We also tested the combination with the TPRV1 agonist capsaicin and the TRPV1 antagonist capsazepin. To date, we found an induction of TPRV1 mRNA by acrolein. 

Figure. mRNA expression of TRPV1 in the human epithelial cell line HT29. Cigarette smoke extract and cigarette smoke components were tested in combination with the TRPV1 antagonist CPZ. Ethanol was included as a control (CPZ is dissolved in ethanol). CSE: cigarette smoke extract. CPZ: capsazepine. Acro: acrolein.

Also, we found that, in the context of an inflammatory environment, IL-8 protein is induced by CS extract, however, we couldn’t unravel yet the role of TRPV1 in this observation.

Figure. Protein expression of IL-8 in the human epithelial cell line HT29. An inflammatory environment was created by adding LPS and IFN-γ in the cell culture medium. Different concentrations of CSE were tested with and without addition of LPS and IFN-γ. IL-8 is induced by CSE in inflammatory conditions. hIL-8 was measured by ELISA. CSE: cigarette smoke extract. L/I: LPS/IFN-γ.

These results were not mentioned in this paper, as in our opinion, these date wouldn’t contribute. 

Minor comments

P9, line 165, the detail information of mice exposed whole body to mainstream CS is necessary. 

The cigarette smoking procedure is described in the Materials & Methods section ‘Cigarette smoke exposure’ and was published by D’hulst et al., 2005 (PMID: 16055867). The same procedure was applied to all mouse cigarette smoke exposure experiments. Line 148-155 reads as: “Mice were exposed whole body to mainstream cigarette smoke, as described previously [28]. Briefly, groups of 10 mice were exposed to the tobacco smoke of five cigarettes (Reference Cigarette 3R4F without filter; University of Kentucky, Lexington, KY, USA) four times a day with 30 min. smoke-free intervals, five days per week for 4 or 24 weeks. An optimal smoke:air ratio of 1:6 was obtained. The control groups were exposed to air. Carboxyhaemoglobin in serum of CS-exposed mice reached a non-toxic level of 8.7 ± 0,31% (compared with 0.65 ±0,25% in air-exposed mice), which is similar to carboxyhaemoglobin blood concentrations of human smokers[29].”

How about gene expressions of other TRPVs, TRPV 2-6? It should be mentioned.

We agree with the reviewer’s suggestion. Next to TRPV1, we also investigated TRPA1. However, no significant results were obtained for TRPA1 in response to cigarette smoke exposure.

Figure. mRNA expression of TRPA1 in ileum, proximal and distal colon of wild-type C57/Bl6 mice after 24 weeks of CS exposure. No significant differences in expression were detected. IL: ileum. PC: proximal colon. DC: distal colon. Air: no cigarette smoke exposure. CS: cigarette smoke exposure of 24 weeks.

---

## [Editor Report · Decision Letter 1]

13 Jul 2020

Translational research into the effects of cigarette smoke on inflammatory mediators and epithelial TRPV1 in Crohn’s disease

PONE-D-20-04633R1

Dear Dr. Allais,

We’re pleased to inform you that your manuscript has been judged scientifically suitable for publication and will be formally accepted for publication once it meets all outstanding technical requirements.

Kind regards,

Mathilde Body-Malapel

Academic Editor

PLOS ONE
---

## [Editor Report · Acceptance letter]

17 Jul 2020

PONE-D-20-04633R1 

Translational research into the effects of cigarette smoke on inflammatory mediators and epithelial TRPV1 in Crohn’s disease 

Dear Dr. Allais:

I'm pleased to inform you that your manuscript has been deemed suitable for publication in PLOS ONE. Congratulations! Your manuscript is now with our production department. 

Kind regards, 

on behalf of

Dr. Mathilde Body-Malapel 

Academic Editor

PLOS ONE